

# Inoculation with *Pseudomonas* spp. in *Solanum lycopersicum* increases yield and fruit quality under nutrient shortage conditions

Patricia Torres-Solórzano[1], Homero Reyes-De la Cruz[1], Josué Altamirano-Hernández[1], Lourdes Macías-Rodríguez[1], Jesús Campos-García[1] and Alfonso Luna-Cruz[2]

[1] Instituto de Investigaciones Químico Biológicas, Universidad Michoacana de San Nicolás de Hidalgo, Morelia, Michoacán, Mexico
[2] Instituto de Investigaciones Químico Biológicas, SECIHTI-Universidad Michoacana de San Nicolás de Hidalgo, Morelia, Michoacán, Mexico

## ABSTRACT

**Background**. Greenhouse tomato cultivation has experienced significant growth in recent years. However, this production system requires high fertilization levels, relying mainly on synthetic agro-inputs. While their use meets the crop's nutritional demand, they present major limitations. Excessive application reduces absorption efficiency, increases soil salinity, and can contaminate water sources. Additionally, rising global fertilizer costs have made it necessary to seek efficient alternatives with lower contamination risks. In this context, plant growth-promoting rhizobacteria (PGPR) are a viable option to reduce inorganic fertilizer use. These microorganisms enhance nutrient availability and stimulate crop development. For this reason, the ability of five *Pseudomonas* strains to reduce nitrogen, phosphorus, and calcium-based fertilizers by up to 50% was evaluated in 'El Cid F1' tomato cultivation under a hydroponic greenhouse production system. Their performance was analyzed both individually and in consortium.

**Methods**. Preliminary *in vitro* tests demonstrated that *Pseudomonas* sp. isolates C13, C14, and C15, *Pseudomonas fluorescens* C30, and *P. putida* ACJ14, both individually and consortium could fix nitrogen, solubilize phosphate and biosynthesize indole-3-acetic acid. Greenhouse trials revealed the potential of the *Pseudomonas* spp. isolates to stimulate vegetative growth while improving fruit quality parameters including firmness, total soluble solids, titratable acidity, and lycopene concentration.

**Results**. The *P. putida* ACJ14 and *Pseudomonas* sp. C14 isolates significantly increased fruit yield by 54% and 73%, respectively. Lycopene content increased to 132.9 mg/kg of fruit with *Pseudomonas* ACJ14 and 130.22 mg/kg of fruit with *Pseudomonas* sp. C14. The consortium showed no significant difference in any parameters compared to individual isolates. All isolates demonstrated rhizosphere persistence for 30 days post-root inoculation.

**Conclusions**. The results position *Pseudomonas* spp. isolates C13, C14, C15, *P. fluorescens* C30, and *P. putida* ACJ14 as sustainable PGPR alternatives enabling 50% reduction of nitrogen, phosphorus, and calcium fertilization in greenhouse tomatoes. The strain *Pseudomonas* sp. C14 emerged as particularly effective, demonstrating

Corresponding author
Alfonso Luna-Cruz,
alfonso.luna@umich.mx

the highest nitrogen fixation capacity, second highest IAA production, and superior performance in seed germination rates, fruit firmness, and lycopene enhancement.

# INTRODUCTION

Tomato (*Solanum lycopersicum* L.) is the world's most widely produced vegetable, with 186 million tons produced in 2022 (*FAO, 2023*). Mexico ranks as the eighth-largest producer and the leading exporter to the international market, contributing 24.7% of the global tomato supply (*SIAP, 2018*). Therefore, due to its significant economic impact, the cultivation of this plant represents one of Mexico's most critical socioeconomic activities in the agricultural sector. In addition, it is considered a substantial food with high nutritional value due to its protein, mineral, and fiber content, and it is also an important source of bioactive compounds with antioxidant activity, including phenols, carotenoids, vitamins, and glycoalkaloids (*Chaudhary et al., 2018*; *Perveen et al., 2015*), health-promoting bioactivities of tomatoes make them useful ingredient for the development of functional foods (*Chaudhary et al., 2018*). It is a critical element in the fight against global malnutrition (*Vats et al., 2020*). It meets nutritional requirements in the pursuit of ensuring food security for the estimated population by 2050 (*Hunter et al., 2017*).

In recent years, there has been a significant increase in tomato production in protected agriculture, primarily in greenhouses, due to its high productivity and higher yields compared to other agricultural systems (*FIRA, 2019*). However, in this type of agriculture, nutritional conditions are provided by applying nutrient solutions that contain a balanced mix of chemical elements, taking into account factors such as the plant's phenological stage, the frequency of application, and environmental conditions. Under these conditions, the deficiency of even one a single nutrient significantly impacts its plant growth and development, ultimately reducing affecting yield (*López-Marín, 2017*). In current greenhouse farming practices, the applied concentration of agricultural inputs often exceeds nutritional requirements to avoid deficiencies, leading to significant loss of inorganic fertilizers through leaching, volatilization, or accumulation in the soil when they exceed the crops' absorption rate (*Burchi et al., 2018*). This results in disturbances to the global ecosystem, including an imbalance in the functions of rhizospheric microorganisms involved in plant growth, development, and nutrition, which limits their agricultural potential and poses a risk to sustainable productivity (*Lenka et al., 2016*).

On the other hand, the irrational use of these products has led to a significant increase in production costs (between 10 and 25%) (*Salgado & Núñez, 2012*). In addition, due to the recent rise in fertilizer prices resulting from various factors (*Organización de las Naciones Unidas para la Alimentación y la Agricultura, 2022*) it is essential to emphasize the importance of adopting agricultural practices that promote the efficient use of natural

resources and minimize fertilizer use. These measures are crucial to ensuring environmental sustainability and preserving ecological balance, securing a healthier future for the planet and its inhabitants (*Kalyanasundaram, Syed & Subburamu, 2021*).

In this sense, research has focused on utilizing microorganisms that inhabit the rhizosphere to promote plant growth, known as plant growth promoting rhizobacteria (PGPR) (*Subrahmanyam et al., 2020*). PGPR constitutes 2–5% of rhizospheric bacteria (*Prasad et al., 2019*). Their presence in the soil improves germination rate and seed vigor (*Rudolph, Labuschagne & Aveling, 2015*), influences root development (*Verbon & Liberman, 2016*), restores soil quality and fertility in contaminated soils (*Kaur, 2021*), protects against phytopathogenic microorganisms and environmental stress conditions (*Shrivastava & Kumar, 2015*; *Kumar et al., 2019*), and helps reduce the global dependence on destabilizing chemical agroinputs (*Laslo & Mara, 2019*). PGPR are involved in transforming nutrients into biologically assimilable forms (*Singh et al., 2024*) or stimulates the development of the root system, leading to an increase in the surface area and proliferation of root hairs, thus improving nutrient acquisition (*Calvo et al., 2016*; *Grover et al., 2021*). Recently, PGPR has been demonstrated as an ecological option that benefits numerous plant species (*Kaur & Reddy, 2015*; *Latef et al., 2020*). Various strains of microorganisms have been isolated, and their inoculation into plants has been shown to positively influence nutrient concentrations, thereby enhancing productivity (*Calvo et al., 2016*). This enables the optimization of inorganic fertilizer application across various crops (*Bona et al., 2017*). *Pseudomonas* is one of the most diverse and widely distributed genera due to its high adaptability in multiple environments (*Poncheewin et al., 2022*). It has an advantage over other bacteria due to its ability to colonize and persist in the rhizosphere (*Zboralski & Filion, 2020*), where it exhibits great metabolic and functional versatility, playing a key role in promoting plant growth or controlling phytopathogenic microorganisms (*Roquigny et al., 2017*). Various activities have been described within this genus, highlighting the production of siderophores (*Kotasthane et al., 2017*) and antifungals (*Hernández-León et al., 2015*). In addition, their ability to fix atmospheric nitrogen (*Fox et al., 2016*), biosynthesize indole-3-acetic acid (IAA) from tryptophan (*Li et al., 2017*), and solubilize phosphates (*Kour et al., 2020*) has been demonstrated. Recently, the presence of antifreeze proteins was reported in three different strains of *Pseudomonas* sp., which are responsible for inhibiting the recrystallization activity of ice. This fact was related to promoting tomato seed germination and root development in plants grown at 14 °C (*Vega-Celedón et al., 2021*), highlighting their adaptability in adverse environments and ability to mitigate abiotic stress.

Moreover, the genus *Pseudomonas* exhibits a wide genetic diversity since only 25 to 35% of the genome of all members of the genus is composed of core genes. At the same time, the rest confer metabolic, ecological, and functional variation at the species level and between strains of the same species (*Loper et al., 2012*). In other words, the performance of bacteria as plant growth promoters varies depending on different factors such as plant genotype, climatic, and growing conditions (soil type, temperature, humidity, *etc.*). In this sense, specific bacterial strains that have shown promising effects under controlled laboratory conditions may fail in real-world situations (*Díaz-Rodríguez et al., 2021*). Therefore, their

agricultural use as bioinoculants remains limited, representing only approximately 1% of the conventional agriculture market. This is despite the market continuing to experience growing demand due to a shifting trend toward sustainable agriculture (*The Insight Partners, 2022*; *The Insight Partners, 2024*; *Mordor Intelligence, 2024a*; *Mordor Intelligence, 2024b*).

In this context, the present research aimed to generate relevant information on the effect of *Pseudomonas* sp. inoculation on the yield and fruit quality of tomatoes grown with reduced macronutrient doses and under greenhouse conditions.

# MATERIALS & METHODS

The research was divided into two phases. The first phase was conducted under laboratory conditions, where the plant growth-promoting traits of the bacterial isolates were characterized. The second phase involved evaluating the bacterial isolates under greenhouse conditions. Information about laboratory evaluation is provided below.

## Evaluation of bacterial activities associated with nutrient availability

In 2018, our research group began isolating native microorganisms from the rhizosphere of commercially essential crops in Michoacán, Mexico, including agave (*Agave* sp.) and sugarcane (*Saccharum officinarum*), to obtain the most outstanding ones in terms of plant growth promotion mechanisms and reduce the application dose of inorganic fertilizers. The best isolates were identified as species of the genus *Pseudomonas*, designated as *Pseudomonas* sp. C13, *Pseudomonas* sp. C14, *Pseudomonas* sp. C15, *P. fluorescens* C30, and *P. putida* ACJ14. The 16S rRNA gene was amplified using the primers 8F (5′-AGAGTTTGATCCTGGCTCAG-3′) (*Edwards et al., 1989*) and 1492R (5′-GGTTACCTTGTTACGACTT-3′) (*Turner et al., 1999*). Sequencing was performed using internal primers: 518F (5′-CCAGCAGCCGCGGTAATACG-3′) and 800R (5′-TACCAGGGTATCTAATCC-3′) (*Ghyselinck et al., 2013*). The five isolates were selected based on their activity related to the transformation of nutrients into biologically assimilable forms based on the following evaluations:

### Biological nitrogen fixation

The ability of the isolates to fix nitrogen was first determined by a qualitative test using a nitrogen-free medium (malic acid: 5.0 g/L; $K_2HPO_4$: 0.5 g/L; $MgSO_4$; 0.2 g/L; NaCl: 0.1 g/L: KOH: 4.5 g/L; $FeSO_4$: 0.5 g/L; bromothymol blue 0.5% on KOH 1N: 2.0 mL/L; $H_3BO_3$: 0.286 g/L; $MnSO_4$: 0.235 g/L; $ZnSO_4$: 0.024 g/L; $CuSO_4$: 0.008 g/L; $Na_2MoO_4$: 0.02 g/L and agar 2.0 g/L (*Dobereiner, Marriel & Nery, 1976*). Subsequently, it was quantified using 200 μL of a bacterial suspension previously adjusted to 0.7 OD (approximately $1 \times 10^9$ CFU/mL) was inoculated into three mL of 10% Soil Extract Broth ($K_2HPO_4$: 0.4 g/L; $MgCl_2$: 0.1 g/L; $MgSO_4$: 0.05 g/L; $FeCl_3$: 0.01 g/L; $CaCl_2$: 0.1 g/L; tryptone: 1.0 g/L; yeast extract: 1.0 g/L; 10% soil extract: 250 mL/L) and incubated with constant shaking at 100 rpm for 72 h. Samples were centrifuged at 6,000 rpm for 15 min, and 10 mL of two M KCl were added. They were then shaken for 60 min and allowed to settle for an additional 60 min. The supernatant was then centrifuged at 6,000 rpm for 10 min. Nitrogen

quantification was performed indirectly by determining the concentration of ammonium ions using the Berthelot colorimetric technique and measuring absorbance in a visible light spectrophotometer at 632.9 nm. The correction of the nitrogen present in the soil extract of the culture medium is carried out using a blank (*Weatherburn, 1967*; *Lara-Mantilla, Villalba-Anaya & Oviedo-Zumaqué, 2007*).

### Quantification of indole-3-acetic acid

The indole-3-acetic acid (IAA) concentration was determined by gas chromatography-mass spectrometry (GC-MS). The isolates were cultured in triplicate for 96 h in 500 mL of Tryptic Soy Broth supplemented with 200 µg/mL of tryptophan. The supernatant obtained after centrifugation was separated and adjusted to pH $5 \pm 0.02$ using 1 N HCl. IAA was extracted with an equal volume of ethyl acetate, evaporated to dryness, and diluted in three mL of HPLC-grade ethyl acetate. It was then dried under a nitrogen gas stream, diluted in two mL of acetyl chloride in methanol (1:4), sonicated for 15 min, and incubated at 75 °C for 1 h. The sample was cooled to 30 °C dried under a nitrogen gas stream, and one mL of dichloromethane and 1.5 mL of acetic anhydride were added. Finally, the derivatized IAA was dissolved in 100 µL of ethyl acetate, and one µL was injected into a GC-MS. IAA quantification was performed using a calibration curve of the IAA standard, derivatized under the same conditions (*Contreras-Cornejo et al., 2009*).

### Phosphate solubilization

Quantification of phosphate solubilized by bacteria in the form of orthophosphates was determined by inoculating 200 µL of the bacterial suspension adjusted to 0.7 OD into 50 mL of NBRIP growth medium (National Botanical Research Institute's Phosphate growth medium (glucose, 10 g: $Ca_3(PO_4)_2$, five g; $MgCl_2 7H_2O$, five g; $MgSO_4 7H_2O$, 0.25 g; KCl, 0.2 g and $(NH_4)_2SO_4$, 0.1 g)) (*Nautiyal, 1999*) and shaking at 150 rpm for 7 days. The sample was centrifuged at 5,000 rpm for 15 min, and the supernatant was used for orthophosphate quantification using the molybdenum blue method (*Murphy & Riley, 1962*). A total of 4 mL of the supernatant was taken, and one mL of the reagent mixture was added. The tubes containing the analyte were incubated at room temperature for 10 min, and the quantification was determined by measuring the absorbance at 882 nm (*Rodríguez-Gámez, Aguilera-Rodríguez & Pérez-Silva, 2013*).

### Calcium solubilization

The quantification of calcium solubilized from calcium carbonate by bacterial activity was determined by atomic absorption spectroscopy. For sample preparation, the bacterial suspension was adjusted to 0.7 OD, and 200 µL was inoculated into 50 mL of culture medium with $CaCO_3$ as an insoluble source of calcium (anhydrous dextrose: 10 g/L; $CaCO_3$: five g/L; $(NH_4)_2SO_4$: 0.5 g/L; NaCl: 0.2 g/L; $MgSO_4$: 0.1 g/L; KCl: 0.2 g/L; yeast extract: 0.5 g/L; $MnSO_4$: 0.1 g/L and agar 15 g/L) followed by incubation for 7 days. The samples were then vacuum filtered with a 0.4 µm membrane, and corresponding dilutions were performed for analysis. Free calcium concentrations were calculated using a standard curve with successive dilutions of calcium carbonate.

## Effect of *Pseudomonas* spp. on germination and vigor of tomato seeds

To evaluate whether bacteria have beneficial effects from seed germination, tomato seeds of the commercial hybrid Saladette El Cid F1 (Harris Moran Mexicana S.A. de C.V.) were disinfected by immersion in sterile distilled water, followed by successive washes with 5.2% sodium hypochlorite for 15 min, 70% ethanol for 3 min, and then again in sterile distilled water. Seeds were inoculated by immersion and shaking at 100 rpm for 1 h in a bacterial suspension adjusted to an optical density (OD) of 0.7. The germination percentage was determined in triplicate by placing 20 previously disinfected and inoculated seeds in Petri dishes with a cotton layer moistened with sterile distilled water, sealing them and keeping them in darkness for 10 days. The percentage of germinated seeds was recorded based on radicle emergence, and the vigor index was calculated using the following equation (*Díaz-Vargas et al., 2001*):

$$Vigor\ index\ (VI) = (M_{rl} + M_{hl}) x GP$$

where: $M$ is the root's mean length (Mrl) and hypocotyl length (Mhl), and GP is the germination percentage.

## Field experiment
### *Experimental design and greenhouse cultivation conditions for the evaluation of Pseudomonas spp. as PGPR*

The second phase of the experiment was conducted in a hydroponic whit use solid substrate (soilless agriculture system) under greenhouse conditions, located at Centro de Bachillerato Tecnológico y Agropecuario No. 7 (CBTA 7, Morelia, Michoacan, Mexico; 19°39′00″N latitude, 101°14′00 E longitude), during the fall of 2021. The average temperature was 25 °C, and light rainfall was recorded during the first ten days of cultivation.

The experimental design consisted of eight treatments (Table 1) with seven replicates, where each tomato plant served as an experimental unit and was placed in a completely randomized design. The tomato plants were arranged in double rows with a spacing of 30 cm between polyethylene black bags and 30 cm between rows. Two rows were assigned to the treatments without fertilizer, two rows to the treatments with 100% fertilization, and two to the treatments with 50% fertilization, where different bacterial isolates were applied. Azoxystrobin (Bankit 25SC) was used at a dose of 400 mL/ha as a preventive phytosanitary control against phytopathogenic fungi.

As part of the treatments, the Steiner Nutrient Solution (*Steiner, 1961*) at 50% nitrogen, phosphorus, and calcium was used for the bacteria-inoculated plants, as well as for control with the same concentration of elements but without bacterial inoculation (T50%) (Table 2). The remaining macro and micronutrients were applied at 100% of the Steiner nutrient solution (*Hoagland & Arnon, 1938*). A control was also treated using the 100% Steiner solution (T100%).

The plants used were obtained from seeds of the commercial hybrid Saladette El Cid F1, which had been previously disinfected and inoculated. The seeds were sown in polystyrene trays with 200 cavities, filled with pre-conditioned peat moss substrate, and placed in an

**Table 1  Greenhouse trial treatments with *Pseudomonas* spp. isolates.**

| Number of treatments | Description |
|---|---|
| 1 | T100% |
| 2 | T50% |
| 3 | Isolate *Pseudomonas* C13 + T50% |
| 4 | Isolate *Pseudomonas* C14 + T50% |
| 5 | Isolate *Pseudomonas* C15 + T50% |
| 6 | Isolate *Pseudomonas* C30 + T50% |
| 7 | Isolate *Pseudomonas* ACJ14 + T50% |
| 8 | Consortium of the five isolates +T50% |

Notes.
T100% and T50% is a treatment at 100% and 50% the Steiner Nutrient Solution (Only N, P and Ca), respectively.

**Table 2  Concentration of macro and micronutrients for the treatments with Steiner nutrient solution.**

| | Nutrients (mmol/L) | | | | | | mg/L | | | | | |
|---|---|---|---|---|---|---|---|---|---|---|---|---|
| | $NO_3^-$ | $H_2PO_4^-$ | $Ca^{2+}$ | $K^+$ | $SO_4^{2-}$ | $Mg2^+$ | $Fe^+$ | $Mn^+$ | $Cu^+$ | $Zn^+$ | $B^+$ | $Mo^+$ |
| T100% | 16.4 | 0.8 | 6.5 | 7.1 | 7 | 4.2 | 0.08 | 0.04 | 0.04 | 0.008 | 0.04 | 0.002 |
| T50% | 8.6 | 0.4 | 3.2 | | | | | | | | | |

Notes.
T100% and T50% is a treatment at 100% and 50% the Steiner Nutrient Solution (Only N, P and Ca), respectively.

area with natural light. Irrigation is applied every 24 h. Transplanting was carried out 40 days after germination when the plants had developed four to five true leaves. Black polyethylene bags (40× 40 cm, 15 L) were used, filled with a mixture of tezontle and soil (100:1); tezontle is considered an inert substrate (*Ojodeagua et al., 2008*), the nutritional characteristics present were: nitrogen 1.42 mg/kg, phosphorus 5.92 mg/kg, potassium 5.64 mg/kg, calcium 7.0 mg/kg, magnesium 2.17 mg/kg, with a cation exchange capacity of 0.097 Cmolc/kg and a particle size of 1–4 mm in diameter. The soil presented the following characteristics: nitrogen, 23.8 mg/kg; phosphorus, 11.20 mg/kg; potassium, 42.57 mg/kg; calcium, 296.17 mg/kg; and magnesium, 4.41 mg/kg, with an organic matter content of 1.23% and a cation exchange capacity of 1.64. This soil is considered nutrient-poor or low in fertility. It is important to note that the soil was used in a low proportion to enhance the colonization of bacterial isolates.

From transplanting to the end of the crop cycle, nutrients were applied using Steiner's nutrient solution (*Steiner, 1961*) at the concentration corresponding to each treatment, with one liter of solution applied per day, distributed in eight irrigations per minute through a drip system. Each plant was inoculated three times during the crop cycle with one mL of the bacterial suspension corresponding to each treatment, adjusted to $1× 10^9$ CFU/mL, and the inoculum was distributed around the base of the stem towards the root zone. The T50% solution treatment plants were inoculated with one mL of sterile culture medium; the T100% control treatment did not receive this application. Inoculations were performed at 10, 70, and 110 days after transplantation. The staking was carried out by guiding the plant using polypropylene raffia and pruning the axillary shoots to allow only the main stem to grow. Since the variety is of indeterminate growth, this was stopped

at the fifth cluster by cutting the apical shoot. Pollination was carried out manually by gently moving the staking raffia when the first floral clusters were formed. Defoliation was performed by cutting the leaves lower than the cluster formed to facilitate illumination and ventilation and avoid excess humidity.

### Analysis of plant response variables and the sensory and physicochemical quality of fruits

The variables related to the vegetative development of each experimental unit were evaluated at 40, 70, 90, and 120 days after sowing. Plant height, stem diameter, and leaf nitrogen content were measured. Leaf nitrogen content was indirectly assessed in SPAD units (Soil Plant Analysis Development) using a SPAD-502 device. It is essential to note that the nitrogen content was not converted to percentage values, as *Xiong et al. (2015)* suggest that it is not always possible to sufficiently control environmental factors to correlate foliar nitrogen percentage with SPAD units.

The fruits were harvested when more than 90% of their surface was red. Firmness was evaluated by making two punctures per fruit at opposite positions located in the equatorial zone using a digital texturometer. The individual weight of each fruit was measured on an analytical balance. The fruits were mixed using an ULTRA-TURRAX homogenizer to obtain puree. Brix degrees were determined by measuring the total soluble solids (TSS) content using a refractometer, where a drop of tomato juice obtained from the previously filtered puree was placed. Titratable acidity was determined by taking 10 mL of the puree, which was then adjusted to 25 mL and neutralized with 0.1 N NaOH using a potentiometer until a pH of 8.2 was reached. Acidity was calculated using the following equation, expressed as a percentage (%) of citric acid (*Sadler & Murphy, 2010*).

$$\% \ acidity = NaOH \ \frac{meq}{ml} VNaOH \ (mL).$$

For lycopene quantification, 0.1 g of tomato puree was weighed and placed in glass tubes covered with aluminum foil. Lycopene was then extracted by adding eight mL of a 2:1:1 (v/v) mixture of hexane, ethanol, and acetone. The tubes were shaken for 10 min, and then one mL of deionized water was added, followed by vortex stirring to induce phase separation. After 10 min of sedimentation, the upper fraction containing the lycopene extracted in hexane was collected. The absorbance of the solution was measured using a spectrophotometer at a wavelength of 503 nm, with hexane as the blank. The concentration of lycopene was estimated using the molar extinction coefficient according to the following equation (*Gawad, Jaiswal & Narayan, 2014*):

$$Lycopene \left( \frac{mg}{kg} \right) = \frac{(A_{503})(537)(10)(0.55)}{(0.1)(172)}$$

where: A503 is the absorbance obtained from the top layer; 537 is the molecular weight of lycopene; 10 is the volume (mL) of the added HEA mixture; 0.55 is the volume of the top layer; 0.1 is the weight of the sample in grams, and 172 is the extinction coefficient of lycopene in hexane.

Finally, yield was determined by quantifying the weight of fruit obtained per plant and calculated as grams of fruit per plant at the fifth cluster.

### Statistical analysis

The data were analyzed using one-way analysis of variance (ANOVA). A Tukey's post hoc test was conducted to compare means and determine significant differences between treatments, with a significance level of $\alpha = 0.05$. All statistical analyses were performed using SAS software, version 9.3 (*SAS Institute Inc, 2012*).

## RESULTS

### The *Pseudomonas* spp. isolates improve the nutrient bioavailability

The five isolates tested were positive for atmospheric nitrogen fixation in the qualitative preliminary *in vitro* test, the establishment of *Pseudomonas* spp. was observed on the surface of the medium, and a color change was also generated due to the presence of the pH indicator bromothymol blue, which presents a blue color due to the increase in pH generated by the products of the fixation process (*Kuan et al., 2016*) (Fig. 1). Quantification revealed different fixation capacities in the five isolates (Table 3). *Azospirillum brasilense* was used as a positive control for this assay since this bacterium has been widely reported as a nitrogen fixer (*Zaidi et al., 2017*). In *A. brasilense*, a fixation of 12.7 µg/mL ammonium was quantified. The five *Pseudomonas* sp. isolates exceeded this value, with the C13 strain fixing 15 µg/mL ammonium, *P. fluorescens* fixing 20 µg/mL, and *Pseudomonas* sp. C15 and *P. putida* ACJ14 exhibited a similar fixing activity, with a concentration of approximately 30 µg/mL *Pseudomonas* sp. C14 exhibited the highest value, 35 µg/mL. On the other hand, the *Pseudomonas* isolates showed the ability to solubilize phosphate and calcium from an insoluble source. A concentration of solubilized phosphate in the form of orthophosphates was quantified in the range of 20 to 90 µg/mL. *Pseudomonas* sp. C14 and *P. putida* ACJ14 exhibited the lowest solubilization values, totaling 20 µg/mL, while *Pseudomonas* sp. C13, *Pseudomonas* sp. C15 and *P. fluorescens* C30 showed values close to 90 µg/mL. Regarding calcium solubilization, the quantitative analysis showed that the C13 isolate had the highest value, with 583.74 µg/mL of soluble calcium. However, atomic absorption spectrometry did not detect soluble calcium for the *Pseudomonas* sp. C14 and C15. Similarly, the *Pseudomonas* sp. isolates exhibited the ability to produce IAA; *P. fluorescens* C30 showed the highest efficiency (12.68 µg/mL), while the other isolates had a phytohormone concentration in the range of 0.061 to 0.840 µg/mL.

### The *Pseudomonas* spp. isolates improve the germination rate and vigor of tomato seeds

The viability of the seeds used in this study was low, with a germination rate of 70% in the control treatment (Table 4). Under these conditions, the germination percentage increased from 8.5 to 18.5% due to *Pseudomonas* sp. inoculation, but without significant differences according to the statistical analysis (Table 3). Despite this, the effect on germination was positively correlated with IAA concentration, as bacteria with the highest production levels of this phytohormone, such as *P. fluorescens* C30 (12.6 µg/mL) and *Pseudomonas* sp. C14 (0.67 µg/mL) and *P. putida* ACJ14 (0.47 µg/mL) generated the most significant increase in germination. In contrast, *Pseudomonas* sp. C15 and *Pseudomonas* sp. C13, which produced the lowest concentrations of IAA, 0.12 µg/mL, and 0.06 µg/mL, respectively, did not exert

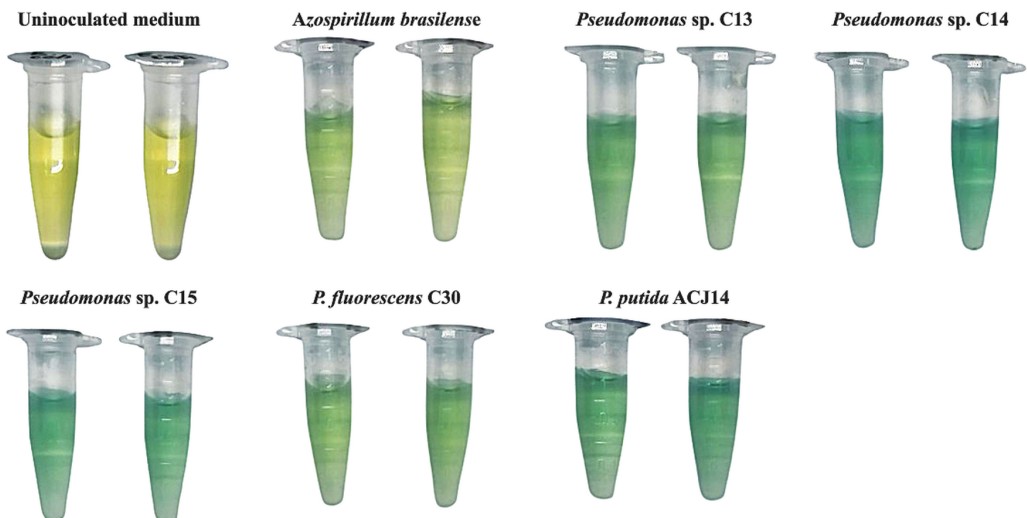

**Figure 1  Quantification showed different fixation capacity in the five *Pseudomonas* spp. strains.** This colorimetric assay qualitatively evaluates nitrogen fixation capacity through blue color development, with increased intensity demonstrating superior activity in the isolate relative to the control.

**Table 3  Isolates activity in attributes associated with nutrient availability.**

| Isolate | Ammonia (mg/mL) | Production IAA (µg/L) | Phosphate solubilization (µg/mL) | Calcium solubilization (µg/mL) |
|---|---|---|---|---|
| *Azospirillum brasilense* sp. 245 | 12.68 ± 0.60 d | 487.33 ± 70.7 b | NA | NA |
| *Pseudomonas* sp. C13 | 15.50 ± 0.9 cd | 61.33 ± 17.2 b | 91.73 ± 1.72 cd | 583.74 ± 4.8 a |
| *Pseudomonas* sp. C14 | 35.07 ± 0.83 a | 840.67 ± 5.8 b | 24.25 ± 0.94 e | – |
| *Pseudomonas* sp. C15 | 30.04 ± 1.6 b | 122.53 ± 5.8 b | 87.75 ± 1.05 d | – |
| *Pseudomonas fluorescens* C30 | 20.45 ± 0.89 c | 12,680 ± 22.9 a | 95.62 ± 2.02 c | 59.42 ± 0.4 c |
| *Pseudomonas putida* ACJ14 | 29.53 ± 1.08 b | 470.67 ± 13.7 b | 21.93 ± 0.17 e | 257.27 ± 2.1b |

Notes.
IIA, Indoleacetic acid.

an evident effect. On the other hand, all *Pseudomonas* sp. isolates significantly increased the plant vigor index. The highest values were recorded for *Pseudomonas* sp. C15, which achieved a 50% increase compared to the control, was showed a 45% increase.

## The *Pseudomonas* spp. isolates improve vegetative parameters in tomato plants with reduced doses of N, P, and Ca

Significant differences were observed in plant height, stem diameter, and SPAD units in leaves (as an indirect measure of leaf nitrogen) at the beginning of the reproductive stage (Fig. 2). The control treatment with 50% Steiner nutrient solution (T50%) showed the lowest plant height, minor stem diameter, and SPAD units compared to the other treatments from the first 90 days after transplanting. However, these differences became more pronounced by 130 days after transplanting. On average, plants under the T50% treatment reached a height of 146 cm, stem diameter of 6.6 mm, and 40 SPAD units, representing a reduction of 22%, 25%, and 18%, respectively, compared to the control

**Table 4** Effect of *Pseudomonas* spp. isolates on germination rate and vigor index of tomato seeds.

| Treatments | Germination (%) | Vigor index |
|---|---|---|
| Control | 70.0 ± 5.7 a | 346.79 ± 10.4 e |
| *Pseudomonas* sp. C13 | 70.0 ± 5.0 a | 409.7 ± 9.5 d |
| *Pseudomonas* sp. C14 | 83.3 ± 4.4 a | 409.74 ± 3.4 d |
| *Pseudomonas* sp. C15 | 76.7 ± 1.6 a | 522.56 ± 2.7 a |
| *Pseudomonas fluorescens* C30 | 81.7 ± 4.4 a | 455.39 ± 13.2cd |
| *Pseudomonas putida* ACJ14 | 80.0 ± 5.0 a | 475.72 ± 8.1 bc |
| Consortium | 80.0 ± 2.88 a | 504.99 ± 13.0 ab |

**Notes.**
The average of three replicates is represented. The bars represent the mean of the standard error. Letters in the same column correspond to means with significant differences (Tukey; $p \leq 0.05$).

plants fertilized with the complete nutrient solution (T100%), which exhibited 177 cm in height, 8.9 mm stem diameter, and 48 SPAD units in the foliage. These results demonstrate that a 50% reduction in the supply of nitrogen, phosphorus, and calcium hurts tomato plant development. The inoculation of plant fertilizer with 50% N, P, and Ca, combined with bacterial isolates had a positive effect on plant height, stem diameter, and SPAD units in the foliage. These effects became particularly evident from the reproductive stage, especially in treatments inoculated with *Pseudomonas* sp. C14, *P. fluorescens* C30, and *P. putida* ACJ14. Inoculation with isolate C14 resulted in the most significant increase in plant height, averaging 194 cm, representing a 9.7% increase compared to T100%. It also resulted in the highest SPAD units values in the foliage, consistent with the *in vitro* assays, as this strain fixed the highest ammonia concentrations.

On the other hand, stem diameter significantly increased with the application of *P. fluorescens* C30 and the bacterial consortium, with plants reaching approximately 10 cm in diameter—13% greater than T100%. These inoculations also contributed to the increase in SPAD units in the foliage, along with the addition of *P. putida* ACJ14. It is important to note that the control with 50% Steiner solution, without bacterial inoculation, exhibited low SPAD values in the foliage. For tomato plants, a foliar nitrogen content between 3% and 5% is considered adequate (*Argerich & Troilo, 2010*), suggesting that nitrogen uptake was improved despite reduced chemical fertilization. However, nitrogen absorption as inferred from SPAD readings—was enhanced by *Pseudomonas* sp. inoculation, with inoculated plants consistently exceeding 3% foliar nitrogen content.

## Analysis of the survival of *Pseudomonas* spp. in the tomato rhizosphere in a hydroponic whit use solid substrate under greenhouse conditions

At the end of the crop cycle, bacterial isolations were determined from soil adhering to the roots of each plant treatment to confirm the persistence of the bacteria. Colonies with morphologies similar to those of the bacteria inoculated in the plants were identified, and their cellular shape was verified using Gram staining and the observation of siderophores production under UV light (*Cezard, Farvacques & Sonnet, 2014*); this, together with prior knowledge of the molecular identification of each bacterial strain. Finally, characteristics

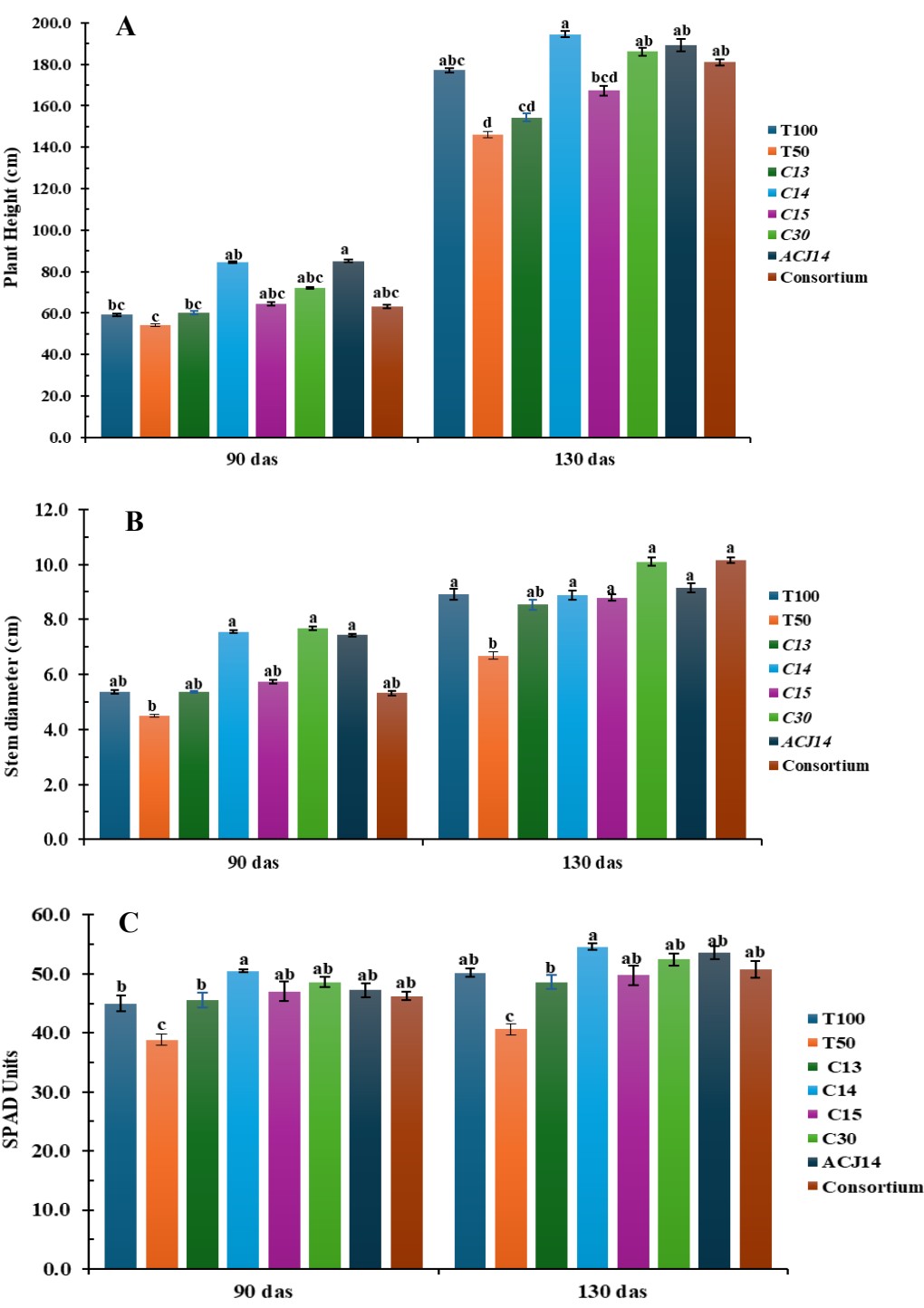

**Figure 2 Tomato growth (reduced fert).** (A) Height (cm), (B) Stem (mm), (C) Chl (SPAD). Mean ± SE ($n = 7$). T100 (100% NPK), T50 (50% NPK), C13-C15, C30 (*P. fluor*), ACJ14 (*P. putida*), Consort. Data are mean ± SE ($n = 7$). Different lowercase letters indicate significant differences among treatments (Tukey's HSD test, $p < 0.05$). Treatments: T100 (100% NPK), T50 (50% NPK), C13-C15 (*Pseudomonas* spp.).

**Table 5** Qualitative tests to evaluate the growth-promoting characteristics of bacteria isolated from previously inoculated soils.

| Treatment | Nitrogen fixation | Phosphate solubilization | Calcium solubilization | IAA production | Fluorescence under UV light |
|---|---|---|---|---|---|
| T100% | − | − | − | − | − |
| T50% | − | − | − | − | − |
| *Pseudomonas* sp. C13 | + | + | + | + | + |
| *Pseudomonas* sp. C14 | + | + | + | + | + |
| *Pseudomonas* sp. C15 | + | + | + | + | + |
| *Pseudomonas fluorescens* C30 | + | + | + | + | + |
| *Pseudomonas putida* ACJ14 | + | + | + | + | + |
| Consortium | + | + | + | + | + |

Notes.
   T100% and T50% is a treatment at 100% and 50% the Steiner Nutrient Solution (Only N, P and Ca), respectively. IAA, Indoleacetic acid.

associated with growth plant promotion were performed (Table 5). Initially, higher bacterial loads were observed in *Pseudomonas* sp.-inoculated treatments compared to non-inoculated controls, with presumptive Pseudomonas colonies isolated from all treatments based on macroscopic morphology; however, control isolates were discarded due to their lack of UV fluorescence and absence of characteristic activities including nitrogen fixation, phosphate/calcium solubilization, and IAA production.

## Effect of inoculation with *Pseudomonas* spp. isolates on the root system of tomato

At the end of the crop cycle, the effect of *Pseudomonas* sp. inoculation on the root system was evaluated by quantifying the dry weight. The decrease in nutrient concentration in T50% plants did not significantly affect root weight compared to T100%, with weights of 8.2 g and 12.7 g, respectively (Fig. 3). The isolates C13, C15, C30, and consortium did not generate significant differences from the controls. On the other hand, inoculation with *Pseudomonas* sp. C14 and *P. putida* ACJ14 resulted in a 63% and 60% increase in root weight, respectively, compared to T100%. Figure 4 illustrates root growth in each of the treatments with bacterial isolates.

## Analysis of the quality of tomato fruit fertilized with 50% N, P, and Ca after inoculation with *Pseudomonas* spp

Tomato fruits were harvested when they were red on more than 90% of their surface. Regarding firmness, the results showed that the 50% reduction of nutrients affected this parameter, with the lowest firmness observed in the T50% treatment, with an average of 4.6 N (Table 6). Likewise, fruit firmness was increased with the complete application of fertilizers (T100%) and bacterial inoculation. On the other hand, there was an increase in TSS due to the application of the complete Steiner solution and bacterial inoculation, compared to the T100% treatment, whose fruits presented 6.2° Brix. The highest TSS was obtained by inoculation with the isolate C14 (7.3° Brix).

On the other hand, the fruits showed values between 0.29 and 0.41% of titratable acidity, with the lowest value being observed in the treatment with 50% fertilization without inoculation and increased significantly due to the application of complete fertilization, as

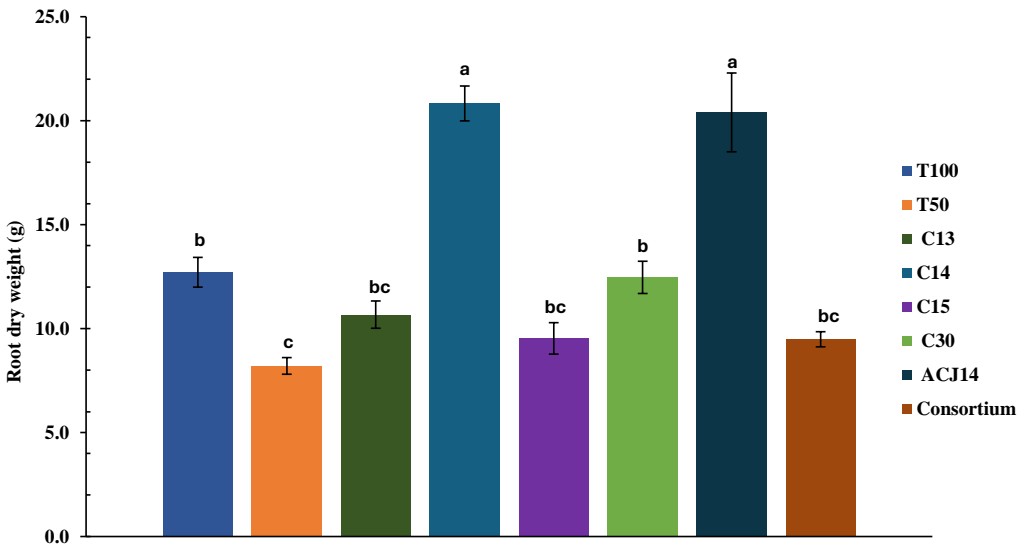

**Figure 3  Tomato root weight (reduced fert).** Mean ± SE ($n = 7$). T100 (100% NPK), T50 (50% NPK), C13-C15 (*Pseudomonas* spp.), C30(*P. fluorescens*), ACJ14 (*P. putida*) and Consortium. Means ± SE ($n = 7$). Different letters show significant differences ($p < 0.05$).

well as by inoculation in the consortium, by 41% in both cases. The rest of the treatments also showed an increase in TSS of up to 37% in the case of the *P. putida* ACJ14 isolate; however, the difference was not statistically significant. Finally, lycopene content was evaluated as a nutraceutical quality parameter. The lycopene content was 94 mg/kg in absolute control fruits (T50%) and was positively affected by bacterial inoculation, with significant increases observed with isolates C14 (37.93%), C15 (37.94%), ACJ14 (40.13%), and the bacterial consortium (30.8%).

## Greenhouse tomato yield under 50% reduced N, P, and Ca fertilization with *Pseudomonas* spp. inoculation

The 50% reduction in nutrients significantly affected fruit production, as the T50% treatment yielded only 933 g of fruit per plant (Fig. 5). It also affected fruit development, resulting in the production of smaller fruit (Fig. 5). Figure 6 illustrates the size of the fruits in each treatment. However, yield increased by 19.4% with complete fertilization, as the total weight of fruit in T100% was 1,115.97 g per plant. Moreover, nutrient deficiency was compensated for by bacterial activity, as inoculation with *Pseudomonas* sp., characterized as PGPR for their stimulation of morphological variables of plant growth, led to an increase in tomato yield under reduced fertilizer conditions, even surpassing the yield generated by conventional agricultural fertilization treatments. Inoculation with the isolates C13 and C15 increased yield compared to T100% treatment by 14.3% and 10.3%, respectively. Inoculation with *P. fluorescens* C30 resulted in a 32.4% increase compared to T100% and a 58.17% increase compared to T50%. The effects generated by *P. putida* ACJ14 and the bacterial consortium were similar, with yields of 1,727 and 1,711 g of tomato per plant, respectively, representing an increase of approximately 54% over T100% in both

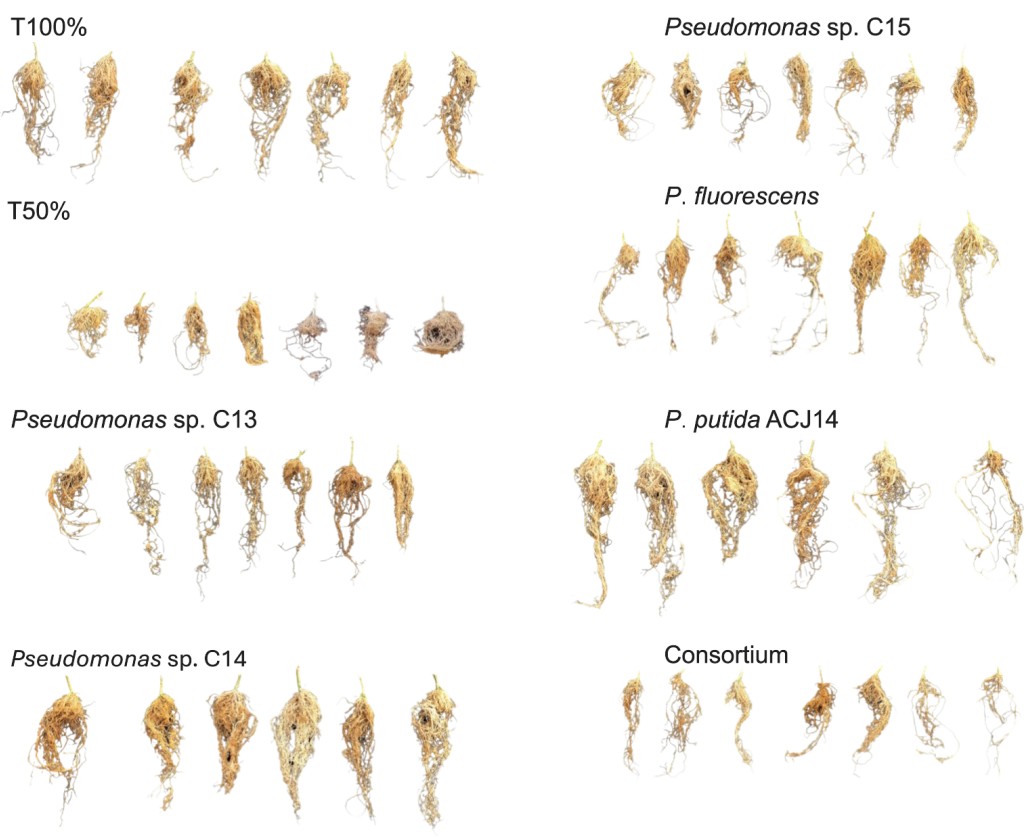

**Figure 4** Illustrative root systems of tomato plants inoculated with different *Pseudomonas* spp. strains under greenhouse conditions.

cases. Unlike the isolates C13, C15, and C30, the yield increase was statistically significant compared to the yield obtained when the complete nutrient concentration was applied. A highly significant increase has been obtained from inoculation with *Pseudomonas* sp. C14, which produced 1,893 g of tomatoes per plant. This yield represents a 102% increase compared to plants fertilized with the reduced dose and a 73% increase compared to the control with the complete Steiner solution. Plants inoculated with this strain exhibited improved growth following transplanting, resulting in the most significant root volume and increased yield and fruit quality.

## DISCUSSION

Fertilizers, mainly phosphates and nitrogen-based, are essential for the global food system and for achieving high agricultural yields (*Erisman et al., 2013*). However, current greenhouses practices significantly exceed the concentration of elements to meet nutritional requirements and prevent deficiencies (*Burchi et al., 2018*). About 55% of nitrogen, 54% of calcium, and 36% of phosphorus are lost before being absorbed by plants due to various factors (*Sanjuan-Delmás et al., 2020*). In this situation, it is feasible to produce crops using a soilless system such as hydroponics. This production method consists of

growing plants without soil, involving the direct supply of the nutrient solution to the plant's root zone, allowing for more efficient use of water and fertilizers compared to other systems (*Touliatos, Dodd & McAinsh, 2016*; *Prakash et al., 2020*); the roots are suspended in a nutrient solution, and therefore, the released compounds are subject to relatively large dilution effects. However, crops grown in a solid substrate equipped with drippers to supply the nutrient solution are less disruptive, as they do not generate the same constant water flow (*Azizoglu et al., 2021*). Additionally, the effect on the rhizosphere extends further in water-based cultivation; however, the concentration of root exudates decreases much more rapidly than in hydroponic systems using a solid substrate (*Raviv, Lieth & Bar-Tal, 2019*). This research evaluated the effectiveness of *Pseudomonas* species isolated from sugarcane in biological nitrogen fixation and the solubilization of phosphate and calcium. Plant growth-promoting strains of *Pseudomonas* spp. are commonly found in species such as *P. fluorescens*, *P. putida*, *P. protegens*, *P. migulae*, and P. *chlororaphis*. However, biocontrol agents can be found across a much broader range of taxa (*Thomas et al., 2024*). Most *Pseudomonas* species lack the coding genes necessary for the biological nitrogen fixation process (*Yan et al., 2008*). However, *Fox et al. (2016)* reported that *P. protegens* is a nitrogen fixer, while *Wang et al. (2017)* reported the ability of *Pseudomonas stutzeri* A1501 to carry out the fixation process under microaerophilic and element-limiting conditions. *P. stutzeri* has been the most studied species of the genus. In this sense, *Ke et al. (2018)* demonstrated an improvement in maize growth and nitrogen availability using the same strain, along with an increase in the population of diazotrophs in the rhizosphere. In addition, it had previously been determined that several Brazilian sugarcane varieties could obtain up to 70% of their nitrogen requirement through this process *Li et al. (2017)*. On the other hand, the genus *Pseudomonas* has been classified as one of the principal producers of organic acids responsible for the solubilization of tricalcium phosphate and rock phosphate (*Vyas & Gulati, 2009*). Therefore, the application of these rhizobacteria can represent a safe and sustainable source of these elements, contributing to the reduction of dependence on inorganic nitrogen, as well as reducing the energy required for its production (*Mahmud et al., 2020*), as it has been estimated that up to 65% of this element supplied to cropping systems by biological nitrogen fixation (*Kuan et al., 2016*). On the other hand, *Gao et al. (2015)* reported the production of 18.47 µg/mL of IAA by *Burkholderia* sp. 7016, which resulted in significant increases in shoot height and weight, root length and weight, and stem diameter in greenhouse tomato plants, as well as an increase in tomato yield in the field.

PGPRs are considered as such, if they have specific characteristics, including increasing the production of auxins (IAA), cytokinins, and gibberellins (*Saleem et al., 2017*; *Vocciantte et al., 2022*) These substances promote increased seedling emergence, enhance vigor and biomass, elevate endogenous phytohormone levels, and improve nutrient use efficiency. In particular, the ability of bacteria to produce IAA constitutes an ecological alternative in sustainable agriculture due to its role in the development of the root system; it promotes plant growth and cell division, as well as root elongation, proliferation, and enhanced branching of root hairs, which increases the root surface area for water and nutrient uptake (*Azizoglu et al., 2021*; *Stegelmeier et al., 2022*).

**Table 6   Effect of *Pseudomonas* spp. isolates on tomato quality parameters.**

| Treatments | Firmness (Newton) | TSS (°Brix) | Titratable acidity (% Citric acid) | Lycopene (mg/kg) |
|---|---|---|---|---|
| T100% | 5.42 ± 0.27 ab | 6.9 ± 0.24 ab | 0.41 ± 0.028 b | 112.85 ab |
| T50% | 4.26 ± 0.15 b | 6.2 ± 0.17 b | 0.29 ± 0.004 b | 94.40 b |
| *Pseudomonas* sp. C13 | 5.15 ± 0.21 ab | 6.6 ± 0.29 ab | 0.39 ± 0.022 ab | 116.74 ab |
| *Pseudomonas* sp. C14 | 5.56 ± 0.25 a | 7.3 ± 0.19 ab | 0.39 ± 0.016 ab | 130.22 a |
| *Pseudomonas* sp. C15 | 4.95 ± 0.16 ab | 6.8 ± 0.21 ab | 0.38 ± 0.017 ab | 130.66 a |
| *P. fluorescens* C30 | 5.30 ± 0.45 ab | 6.9 ± 0.3 ab | 0.39 ± 0.047 ab | 116.87 ab |
| *P. putida* ACJ14 | 5.20 ± 0.34 ab | 7.0 ± 0.22 ab | 0.40 ± 0.027 ab | 132.29 a |
| Consortium | 5.46 ± 0.17 ab | 7.1 ± 0.29 ab | 0.41 ± 0.008 a | 123.53 a |

**Notes.**

TSS, Total soluble solids. T100% and T50% is a treatment at 100% and 50% the Steiner Nutrient Solution (only N, P and Ca), respectively.

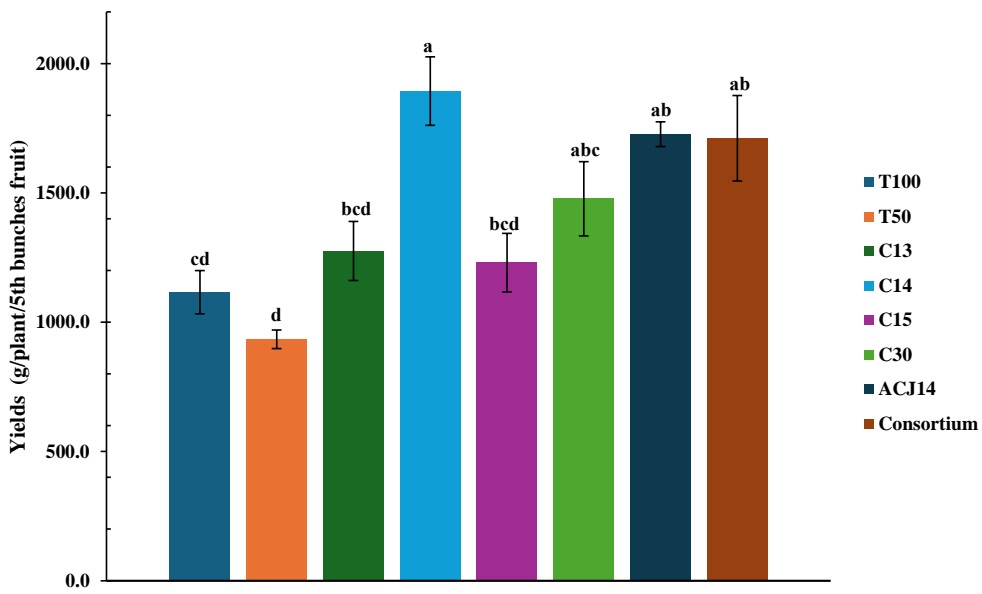

**Figure 5   Tomato yield under *Pseudomonas* spp. inoculation.** Mean ± SE ($n = 7$). T100 (100% NPK), T50 (50% NPK), C13-C15 (*Pseudomonas*), C30 (*P. fluorescens*), ACJ14 (*P. putida*), Consortium. Error bars represent mean ± standard error ($n = 7$). Different lowercase letters indicate statistically significant differences (Tukey's HSD test, $p < 0.05$).

From another perspective, the production of IAA by the evaluated *Pseudomonas* strains may be associated with their ability to enhance the percentage of germination and vigor of tomato seeds, particularly during seedling emergence, by stimulating cellular elongation of the embryonic axis (*Li et al., 2016*).

Balanced nutrient intake is essential at all stages of plant growth. For greenhouse tomatoes, *Bodale et al. (2021)* identified the first 90 days of growth as the period of highest nutrient consumption, mainly nitrogen, phosphorus, and potassium, due to their involvement in stem growth and leaf and flower formation. Thus, plant growth is limited
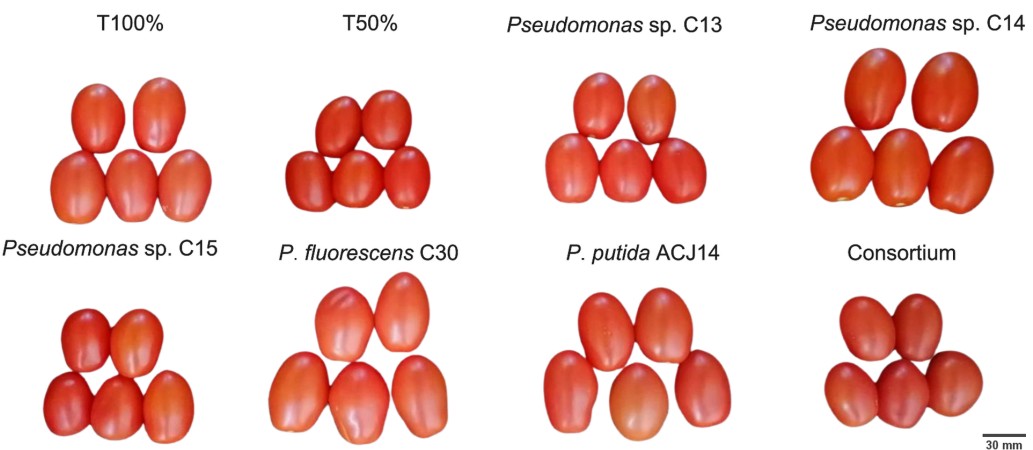

**Figure 6** **Illustration of tomato fruits harvested from plants inoculated and non-inoculated with *Pseudomonas* spp.** Tomato fruits from different treatments inoculated with *Pseudomonas* spp. isolates. Images are representative of observed morphological differences.

when nutrients are scarce, and crop yields are consequently reduced. Determining the activities related to the increase in nutrient availability by the isolated *Pseudomonas* allowed for a 50% reduction in the concentration of N, P, and Ca in the Steiner nutrient solution. Scientific evidence suggests that *Pseudomonas* species act as plant growth-promoting agents in hydroponics, enhancing various biological parameters in plants, even under optimal growing conditions (*Etesami & Glick, 2024*). For instance, inoculating hydroponically grown lettuce in a nutrient film technique (NFT) system with *P. lundensis* and *P. migulae* resulted in significant improvements in several key agronomic parameters, including a 55.3% increase in shoot fresh weight, as well as increases in leaf number, leaf area, and both fresh and dry root weight (*Mei et al., 2023*). A higher accumulation of IAA in the leaves and roots of inoculated plants is attributed to the production of this phytohormone by *Pseudomonas* species (*Etesami & Glick, 2024*). *Lee et al. (2010)* found that the *Pseudomonas* sp. strain LSW25R promoted growth in Momotero tomato plants by successfully colonizing the rhizosphere in a hydroponic system under nitrogen (N) and calcium (Ca) deficient conditions. These plants showed increased height and fresh weight when treated with $10^8$ CFU ml$^{-1}$ of this strain, suggesting that, in addition to IAA production, various mechanisms contribute to growth. Additionally, using hydroponic systems for *Romaine lettuce*, reducing the nutrient concentration by 50%, and inoculating with a PGPR consortium, including *P. fluorescens*, significantly improved productivity and macronutrient uptake (*Aini, Yamika & Ulum, 2019b*).

In our experiment, data obtained from measurements of height, stem diameter, and leaf nitrogen content indicated a nutritional compensation in tomato plants with reduced fertilization generated by inoculation with *Pseudomonas* sp., probably due to their participation in facilitating the availability of essential macronutrients at these stages through their ability to fix nitrogen and solubilize phosphate, or through the action of substances synthesized by the bacteria that stimulated plant elongation and nutrient uptake,

improving root development. Consistently, bacteria with greater capacity to perform the activities above showed an enhanced effect on tomato growth. These results showed a clear pattern of nutrient deficiency in uninoculated plants when the nitrogen, phosphorus, and calcium concentration in the nutrient solution was reduced by 50%. Under these conditions, plants exhibited reduced stem diameter, shorter height, lower leaf nitrogen percentage, and smaller root volume, indicating that nutrient reduction can jeopardize crop yield by affecting plant development. However, the improvement in the variables measured in the inoculated plants suggests that the *Pseudomonas* sp. isolated from sugarcane and agave contributed significant amounts of these elements to the crop through nitrogen fixation and phosphate and calcium solubilization, allowing a better plant development, even more than fertilization with the recommended dose of Steiner solution (*Steiner, 1961*).

Furthermore, the genus *Pseudomonas* has an additional advantage for root colonization due to its ability to form biofilms, which favors its persistence (*Zboralski & Filion, 2020*). The isolates in this study demonstrated the ability to establish themselves near the tomato root and a high probability of surviving in the rhizosphere under uncontrolled conditions for at least 30 days, considering the last inoculation. Additionally, it was confirmed that they maintain their ability to fix nitrogen, solubilize phosphate and calcium, and produce indoles.

Proper nutrition is essential for plant development throughout the life cycle, so an imbalance can affect both vegetative and reproductive growth, resulting in reduced yield and quality (*Bodale et al., 2021*). Firmness is an important parameter related to the stage of a fruit's ripeness; as ripeness increases, firmness decreases due to the hydrolysis of starches and pectins or the degradation of cell walls (*Cárdenas-Coronel et al., 2012*). An optimal calcium application also provides greater cell wall rigidity, directly influencing fruit firmness (*Ahmed et al., 2022*). This could explain the low firmness levels observed in the treatment with reduced fertilization (T50%). However, fruit firmness also depends on other factors. For example, *Mena-Violante & Olalde-Portugal (2007)* obtained a similar value for fruit firmness in a control treatment (4.97 N) and an increase of 1.11 units with *Bacillus subtilis* BEB-13BS inoculation, which they associated with a decrease in the activity of the polygalacturonase enzyme responsible for the degradation of polyuronide, a component of the tomato cell wall. On the other hand, the role of ethylene in the ripening process is well-known (*Zhao, Nakano & Iwasaki, 2021*). Furthermore, the expression of the enzyme 1-aminocyclopropane-1-carboxylate (ACC) deaminase by *Pseudomonas* sp. bacteria has been widely described as a growth-promoting mechanism (*Glick & Nascimento, 2021*), which could modulate ethylene synthesis, slowing down the ripening process and thus reducing the loss of firmness. This parameter is crucial in fruits, as it provides resistance to pathogen attack or damage caused by product handling. Therefore, the results suggest that inoculation with *Pseudomonas* sp. could contribute to reducing tomato postharvest loss and prolonging its shelf life by improving fruit firmness. TSS and titratable acidity determine the flavor of tomatoes (*Gómez y Camelo, 2002*). The content of TSS, measured as °Brix, depends on the degree of ripeness and the tomato variety (*Katsenios et al., 2021*). Since most TSS are composed of sugars, the observed increase in TSS could be related to higher photosynthetic efficiency. *Hernández et al. (2020)* observed a significant

increase in sugar concentration in fruits grown with lower nitrogen concentrations because nitrogen limitation reduces vegetative growth, resulting in greater light irradiation to the fruit, which improves its photosynthetic activity and, consequently, its sugar content. In this study, vegetative growth was not affected by nitrogen reduction in the inoculated treatments; however, removing the lower leaves at fruit ripening was performed, allowing the conditions mentioned above to be met. In the same vein, *Aini, Yamika & Pahlevi (2019a)* correlated the increase in nutrient concentration to sugar content. Specifically, the increase in nitrogen, a basic component of chlorophyll, would be correlated to the increase of photoassimilates, which could explain the increase in leaf nitrogen content observed following inoculation with *Pseudomonas* sp. On the other hand, TSS content is directly related to the degree of ripeness and, consequently, to the loss of firmness (*Kaur et al., 2006*). However, the values obtained in each of the treatments follow the same trend determined by firmness; that is, the treatments with higher firmness had higher TSS content, which translates into a high capacity for mechanical resistance. This indicates that inoculation with growth-promoting *Pseudomonas* sp. significantly improved two factors that define tomato quality. Finally, lycopene content was evaluated as a nutraceutical quality parameter, being the most important bioactive component in tomatoes due to its antioxidant capacity (*Assar et al., 2016*). Lycopene content increased due to bacterial inoculation. Previous studies have analyzed the role of microorganisms in modulating bioactive compounds in tomatoes (*González-Rodríguez et al., 2018*; *Singh & Pandey, 2021*), although contradictory results have been reported. For example, *Inculet et al. (2019)* observed an increase in lycopene concentration in tomatoes inoculated with PGPR, attributing it to the improvement of the plant's nutritional status. However, *Katsenios et al. (2021)* analyzed the effect of *Pseudomonas* sp. 19Fv1T and *P. fluorescens* C7 on 70% fertilized tomatoes, where they ruled out an alteration in lycopene content in tomato fruits due to nutrient limitation. In the same context, *Hernández et al. (2020)*, established a correlation between lycopene content and nitrogen application at different phenological stages. The researchers observed greater lycopene accumulation in tomatoes receiving high nitrogen concentrations after transplanting. However, when nitrogen availability was restricted during flowering, lycopene levels increased due to reduced vegetative growth, which enhanced fruit light exposure and stimulated carotenoid synthesis (*Hernández et al., 2020*). From another perspective, the increase in lycopene in tomato fruits was related to the ability of PGPR to reduce the negative effects caused by stress, leading to the production of antioxidant molecules (*Dorais, Ehret & Papadopoulos, 2008*; *González-Rodríguez et al., 2018*). Therefore, the increase in lycopene content observed after inoculation with *Pseudomonas* sp. can be explained by the improvement in leaf nitrogen content during the early stages of plant development, facilitated by the nutrient availability favored by the bacteria, together with leaf removal practices in later stages, which leads to increased light irradiance reaching the fruit. However, lycopene accumulation depends on several biotic and abiotic factors, not on a direct response to the bacteria (*De la Osa et al., 2021*).

According to the *in vivo* trial results, inoculating plants with *Pseudomonas* sp. increased the morphological parameters evaluated in those fertilized with 50% nitrogen, phosphorus,

and calcium from the Steiner nutrient solution, confirming its ability to promote plant growth. Additionally, the fruits maintained their firmness, high sugar concentration, titratable acidity, and lycopene content, surpassing those of the conventional treatment.

In agricultural practice, achieving high yields is the primary goal of producers. Authors such as *Aini, Yamika & Pahlevi (2019a)* and *Aini, Yamika & Ulum (2019b)* point out a direct relationship between nutrient concentration, vegetative development, and crop yield. Tomato cultivation requires large amounts of nitrogen, phosphorus, and potassium, so a deficiency in any of these elements severely affects its growth and yield (*Zaidi et al., 2017*), as observed in the T50% treatment, where the reduction in nutrient concentration caused a decrease in fruit yield, which was increased by inoculation with *Pseudomonas* sp. According to *Oteino et al. (2015)*, current research suggests that inoculating with phosphate-solubilizing bacteria may reduce up to 50% the application of phosphate without significantly reducing yield. This study demonstrated that inoculation with *Pseudomonas* sp. exhibiting characteristics associated with plant growth promotion, including phosphate solubilization, allows for a 50% reduction in application rates of phosphorus, nitrogen, and calcium, resulting in a significant increase in tomato yield and the size of harvested fruits. The tomatoes were classified as marketable fruits according to the Mexican standard NMX-FF-031-1997-SCFI Productos Alimenticios No Industrializados Para Consumo Humano—Hortalizas Frescas—Tomate—(*Lycopersicun esculentum* Mill.)—Especificaciones (*NMX-FF-031-1997-SCFI, 1997*), which establishes that their equatorial diameter must be greater than 38 mm (*Norma Mexicana NMX-FF-031-1997-SCFI, 2023*).

Several authors observed similar effects following inoculation with *Pseudomonas* sp. on different solanaceous crops. For instance, *Chiquito-Contreras et al. (2017)* evaluated the impact of three isolates of *P. putida* individually and in combination on the growth and productivity of habanero peppers in a greenhouse, applying complete and 25% reduced inorganic fertilization. They observed significant increases in yield, particularly when plants were fertilized with a 75% nutrient solution, which reduced production costs and environmental pollution while achieving a productivity increase up to 36%. However, no differences were observed in inoculation combined with complete fertilization; they argue that the efficacy of PGPR is particularly notable at reduced fertilizer doses. This suggests that the action of these plant growth-promoting microorganisms is especially significant when nutritional elements are scarce in the rhizosphere. Similar results were reported by *Batool & Altaf (2017)*, who found no positive effects on bell pepper yield with PGPR inoculation when conventional nutrition was applied. In addition, based on the yields obtained, it was determined that inoculation allows for a reduction of between 75%–80% in fertilizer concentration, achieving yields statistically equivalent to those of plants with traditional inorganic fertilization. However, doses below 75% significantly affect productivity. Thus, despite the improved efficiency in plants due to the inoculation of beneficial microorganisms, their use does not replace inorganic fertilizers but complements them. The involvement of bacteria in yield enhancement has been associated with various activities. *Narendra et al. (2015)* proposed that the beneficial effects of PGPR are attributed to enhanced nutrient uptake, which is facilitated by their ability to

solubilize phosphorus and fix nitrogen. Specifically, improved phosphorus uptake has been related to its participation in the flowering process. It could provide a higher number of fruits per plant, possibly through its role in root development (*Coutinho-Edson et al., 2014*). *Pérez-Rodríguez et al. (2020)*, in fact, attributed the increased yield in tomatoes to the higher development of the root apparatus resulting from inoculation with *P. fluorescens*, which in turn enhanced water and nutrient uptake ability. Meanwhile, *Espinosa-Palomeque et al. (2019)* attributed the observed increase in tomato yield to an improvement in chlorophyll content, which led to higher photosynthetic efficiency and, subsequently, an increase in tomato weight. In this case, the growth promotion and yield improvement observed in plants inoculated with *Pseudomonas* sp. and fertilized under a reduced nutrient regime could be attributed to increased nutrient uptake and improved nutritional status due to their demonstrated ability to fix nitrogen, solubilize phosphate, solubilize calcium, and produce IAA, as observed in leaf nitrogen content. Additionally, improved root system development may have contributed to these observed effects.

Notably, the isolate *Pseudomonas* sp. C14 demonstrated superior nitrogen fixation capacity and was the second-highest producer of indole-3-acetic acid (IAA). Upon inoculation in tomato plants, this strain was associated with the highest seed germination rate, increased fruit firmness, and elevated lycopene content in the fruit. According to *Rojas-Solís, Hernández-Pacheco & Santoyo (2016)*, not all combinations or consortia of bacterial strains are necessarily effective in promoting plant growth in *Physalis ixocarpa*; their study specifically evaluated various *Pseudomonas* strain combinations, some of which resulted in the inhibition of growth-promoting effects. Similarly, *Kang et al. (2014)* reported that while *Bacillus pumilus* WP8 and *Erwinia persicinus* RA2 are individually effective in promoting tomato plant growth, their combined application not only failed to enhance growth but also negatively affected biocontrol activities.

## CONCLUSIONS

Inoculation with *Pseudomonas* spp. resulted in a 50% reduction in the application of nitrogen, phosphorus, and calcium fertilizers in tomatoes grown in greenhouses using hydroponic irrigation with the Steiner Universal Solution. This increases tomato fruit yield by up to 73% while maintaining or improving the quality parameters of the plants fertilized with the conventionally recommended dose. When applying the isolates in a consortium, it was not observed that they stood out compared to the use individually. The finding demonstrates the potential for utilizing this PGPR approach as a viable alternative to optimize chemical fertilizers, resulting in a notable decrease in the use of environmentally detrimental chemical fertilizers. The isolates *Pseudomonas* sp. C14 stood out for fixing the highest amount of nitrogen and being the second-best producer of IAA. When inoculated into tomato plants, it promoted the highest seed germination percentage, increased fruit firmness, and enhanced its lycopene content.

### Funding

Patricia Torres-Solórzano's master's degree was supported by a scholarship from the Secretaría de Ciencia, Humanidades, Tecnología e Innovación (SECIHTI, Mexico). The funders had no role in study design, data collection and analysis, decision to publish, or preparation of the manuscript.

### Grant Disclosures

The following grant information was disclosed by the authors:
The Secretaría de Ciencia, Humanidades, Tecnología e Innovación (SECIHTI, Mexico).

### Competing Interests

Jesús Campos-García is an Academic Editor for PeerJ.

### Author Contributions

- Patricia Torres-Solórzano conceived and designed the experiments, performed the experiments, analyzed the data, prepared figures and/or tables, authored or reviewed drafts of the article, and approved the final draft.
- Homero Reyes-De la Cruz conceived and designed the experiments, authored or reviewed drafts of the article, and approved the final draft.
- Josué Altamirano-Hernández conceived and designed the experiments, authored or reviewed drafts of the article, and approved the final draft.
- Lourdes Macías-Rodríguez conceived and designed the experiments, authored or reviewed drafts of the article, and approved the final draft.
- Jesús Campos-García conceived and designed the experiments, authored or reviewed drafts of the article, and approved the final draft.
- Alfonso Luna-Cruz conceived and designed the experiments, performed the experiments, analyzed the data, prepared figures and/or tables, authored or reviewed drafts of the article, and approved the final draft.

### DNA Deposition

The following information was supplied regarding the deposition of DNA sequences:
The group *Pseudomonas* spp. sequences are available at GenBank: PV549323 to PV549327.

Isolates available at:
- Isolate C13: https://www.ncbi.nlm.nih.gov/nuccore/PV549325
Accesion: PV549325
- Isolate C14: https://www.ncbi.nlm.nih.gov/nuccore/PV549323
Accesion: PV549323
- Isolate C15: https://www.ncbi.nlm.nih.gov/nuccore/PV549326
Accesion: PV549326
- Isolate C30: https://www.ncbi.nlm.nih.gov/nuccore/PV549324
Accesion: PV549324

- Isolate ACJ14: https://www.ncbi.nlm.nih.gov/nuccore/PV549327
Accesion: PV549327

## Data Availability

The raw measurements are available in the Supplementary Files.

## Supplemental Information

Supplemental information for this article can be found online at http://dx.doi.org/10.7717/peerj.19796#supplemental-information.

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
