# Peer review of "Inoculation with Pseudomonas spp. in Solanum lycopersicum increases yield and fruit quality under nutrient shortage conditions"

_PeerJ, doi:10.7717/peerj.19796_

## Round 0.1 · original submission · Major Revisions

The MS is overall scientifically convincing. The reviewers suggest a number of improvements. Therefore, the authors should consider all the points raised by the reviewers, changing the text accordingly, or explaining why they cannot accept the suggested change. I have also included an annotated pdf with some changes to improve the title and figure clarity.

Reviewer 1 ·

Basic reporting

Although the article is clearly written, I would recommend the assistance of a native speaker to improve sentence construction.
The article include sufficient introduction and background matching the fild of interest. The choice of bibliography is good, but some parts need to be implemented. The structure conforms to the journal's standards and the high-quality figures are well-functioning for understanding the text.
The requests for revision are listed below.
L48
This 2016 production data could be updated and/or provided by a more authoritative source, for example, worldwide study organizations such as I found:
The global production of vegetables went up by 71 percent since 2000, 1.17 billion tonnes in 2022. Tomatoes ranked as the most produced vegetable with 186 million tonnes in 2022.
Source: FAO. 2023. Agricultural production statistics 2000–2022. FAOSTAT Analytical Briefs, No. 79. Rome.https://doi.org/10.4060/cc9205en

L61 -65
Whether the nutrient solutions contain chemical elements balanced according to phenological status, frequency of application, environmental conditions, etc. The following sentences should be better bound (line 64 65) For example by writing: Under these conditions, the deficiency of even one nutrient limits plant growth and development, affecting yield (López-Marin, 2017). Therefore, in the current practice (referred to where?), the applied concentration of agricultural inputs exceeds the nutritional requirements to avoid deficiencies;.... Contextualise where this practice is common
L73 -75
“In addition, the recent rise in the price of fertilizer due to various factors requires implementing agricultural production techniques focused on the efficient use of resources, with a trend towards sustainable agriculture (FAO, 2012)"
The implementation of resource-efficient (sustainable) agricultural production techniques should result from the pursuit of environmental protection and not from increasing fertilizer costs. Sustainable agriculture should be a goal not a trend”.
L78-95
Strengthen the bibliography to support claims about PGPRs (which is very rich on the subject).

Experimental design

The research question is well defined and fills an identified gap in knowledge, but some clarification would be advisable, about methodology too.

L127
For a better understanding of the text, line 127 should be a sub-paragraph of line 120, as well as 138, 152 and 163.

L 172
Can you clarify whether what is described is a preliminary test, something from the bibliography or a proof of the research? If it is the latter, the test should be anticipated in the introduction, where the whole research is presented and it is explained that there are two tests (one on seed germination and the other on greenhouse plants). Furthermore, further on (L 194) a Saladette El Cid F1 hybrid is mentioned, are the Saladette's also from this trial?

L186
If this is a second trial, it could have a title like the first ‘Effect of ...’. ‘Evaluatin of’ in tomato ....
cultivation.

L194
After talking about plants in experimental design, the seeds from which these plants are obtained are discussed; it would be better to present the two stages of the experiment in chronological order.

L196
“…placed in an area with natural light, with irrigation applied every 24 h” It should be specified in what kind of environment this research phase will take place. If the light is natural, what is the season, what is the country, what is the latitude?

Validity of the findings

The results are very interesting and enrich the pool of knowledge on the use and outcomes of PGPR in greenhouse tomato cultivation, and can make a contribution to environmental protection. Something remains to be clarified for a better presentation of the work.

L 262/269
By what method was nitrogen fixation by Azospirillum quantified? And the ability to solubilize phosphate and calcium?

L 352
The first time the acronym SSC is used, it must be accompanied by its full name (Soluble Solids Concentration).

L 365-367
Il riferimento alle figure non è corretto. In Fig 3 c’è la massa radicale e in Fig 4 c’è il grafico Con i dati produttivi espressi in g.

L 379
Replace with "A highly significant increase has been obtained from inoculation with Pseudomonas sp. C14, which produced 1,893 g of tomatoes per plant".

L 568
Indicate the origin of the material at the beginning of the article and not in the conclusions.

L575
If you are going to give a talk about saving money by limiting the use of chemical formulations, you should give the cost of using biostimulants as a comparison. If these are higher than or equal to the cost of chemicals, explain that they may become less expensive as knowledge spreads. Otherwise, emphasise the environmental aspect.

Additional comments

no comment

Reviewer 2 ·

Basic reporting

The present study investigates the effects of inoculation with different Pseudomonas strains on yield, plant fitness and fruit characteristics of tomato plants grown on a substrate consists of tezontle and soil. Overall, the authors present an experiment, with the appropriate controls and I enjoyed reading their manuscript. Following I have some comments and points that need clarification and some suggestions that will help the authors to present their data in a more uniform way without losing the essence of their results.

Tomato plants grew in bags that contained a mixture of tezontle (a pozzolan, a volcanic rock) and soil, and is not clear in the whole manuscript for what cultivation system they talk about, is it hydroponics, is it soil? This must be clearly described to understand what type of fertilization is relevant to test for. For example, Steiner nutrient solution is relevant for a hydroponic system but not for soil, where granular or slow-release fertilizers are applied but for hydroponic systems nutrient losses differ compare to soil (e.g. L 392). Moreover, solubilization is a relevant mechanism for soil environment, where nutrients are bound in Ca+ in clays (e.g. L 392) and how it is relevant to the substrate that the authors tested? In L 424 authors refer to nutrient availability by microbial inocula, which is true for soil systems, how is this applicable to tezontle and soilless systems, where nutrients are applied with solutions in plant-available forms? It would be informative to report on tezontle and sol characteristics.

Experimental design

Concerning Material and Methods Section it would be of benefit for the readers the authors to provide more information with regard to the experimental design, like describe randomization of the experiment, provide a sketch for their treatments, how replications were distributed in space?
L127 It is not clear how this method can distinguish if the measured ammonium comes from N mineralization OR from N fixation OR is just the ammonium that is fixed in soil clays. Usually researchers determine N-fixing ability by using N-free medium, eg
Kuan et al. PLoS One. 2016 doi:10.1371/journal.pone.0152478; Tang et al. Microorganisms 2020 https://doi.org/10.3390/microorganisms8030442
L130 Please provide recipe or reference for the recipe.
L163 To my knowledge the adopted method here with the Pikovskaya medium is mainly applied to determine P - solubilization, while for Ca calcium carbonate (CaCo3) as an insoluble calcium source is used, check on Ahmad et al, Ann Microbiol 73, 34 (2023). https://doi.org/10.1186/s13213-023-01736-5
When I searched for the citation Paguay y Vasco, 2013 I found that it is a Master thesis that describes the exact recipe for the modified medium, that includes both tri-Ca-P and
CaCO3. For clarity and better method presentation and of course reproduction from other researcher, I suggest to describe the full recipe in the text. Moreover, I am unaware if a thesis is a valid citation, if it is please provide the url of the full text as well.
L221 The relationship between SPAD readings and leaf N content is affected by environmental factors and leaf features. Xiong, D., et al. Sci Rep (2015). https://doi.org/10.1038/srep13389. It is valid to determine the relationship (model,regression analysis) between SPAD with some real N% measurements for each experiment and then apply this specific relationship to the whole dataset. SPAD is usually highly correlated to chlorophyll content and greenness, so if not possible to measure %N it is recommended to demonstrate SPAD measurements as SPAD
measurement and not to extrapolate to %N content.
L233 meq/ml it is better to use SI units
L254 For clarity is better to demonstrate the two factors that you compare, for example Y~ Factor1*Factror2. Then post-hoc analysis is performed to determine with levels within a Factor are different

Validity of the findings

My comment that refers to the whole results section, the authors should follow a uniform way to describe and compare their treatments. Which treatment they consider as control? I imagine full fertilization (100%). How application of half of the nutrients (50%) affected the measured parameters and whether and how inoculation reversed this effect. Is there a specific inocula that outperformed for all measured parameters or did each inoculant favored different aspects of plant fitness? The authors have to answer based on their results whether substituting fertilizers with microbial biostimulants favored or not, production, nutrition and plant fitness.
L260 Please clarify that you mention results for in vitro analysis
L261-263 This is materials & methods information
L285-288 Why you report IAA in μg/ml while in the rest of the manuscript you report μg/L? Please choose a uniform way for clarity.
L321 Unfortunately, colonies morphologies are not enough to recognize a bacteria and fluorescence under UV light is a characteristic for P. fluorescence only. Other more sophisticated methods are required for detecting the application of an inoculum, like molecular detection with specific primers, microscopic measurements after GFPl labeling of the strain ect.. Especially when you inoculate a non-sterile system.
L349-352 This sentence is not easy to understand

Additional comments

Table 1 For clarity and uniform presentation you can replace T with SNS

Figures

It would be nice, if the authors apply the same color palette to all figures. Secondarily, in the figures T100% should be demonstrated first as it is the control treatment that represents current application.
In Figure 2 post-hoc letters are not seemed correct. Can please the authors confirm?
Figure 3 and 5 are not informative because they lack a relevant and proper scale for comparing treatments. Now they are just pictures of roots and fruits and you don’t know how much the treatments differ.

Reviewer 3 ·

Basic reporting

The study entitled ‘Inoculation of Solanum lycopersicum L. with Pseudomonas sp. as plant growth promoters and reduced doses of macronutrients increases the yield and quality of tomato fruits’ written by Patricia Torres-Solórzano, Homero Reyes-de la Cruz, Josué Altamirano-Hernández, Lourdes Macías-Rodríguez, Jesús Campos-García, and Alfonso Luna-Cruz describes the effect of Pseudomonas sp. on inoculated tomato cultivars in terms of plant growth and nutrient availability. The manuscript is clear and unambiguous with minor issues that need to be addressed to improve the quality and readability of the work.

Major Issues:
#1
Please consider rephrasing the title of the manuscript. The title ‘Inoculation of Solanum lycopersicum L. with Pseudomonas sp. as plant growth promoters and reduced doses of macronutrients increases the yield and quality of tomato fruits’ is too wordy. The first sentence is not in parallel with the second sentence structure.

Minor issues to be addressed:
-The nomenclature of Pseudomonas sp. maybe miswritten. I am not expert on the subject but after a quick search, I found that ‘The designation “sp.” after a genus refers to a single unnamed species, while the designation “spp.” after a genus refers to more than one unnamed species. Example: Salmonella spp. refers to more than one species of Salmonella.’ Hence, please consider rechecking each time you prefer to use Pseudomonas sp. is consistent with correct abbreviation.
-On Line 52, the sentence ending with ‘which attributes to it the potential to be used as a basis in the development of functional foods (Chaudhary et al., 2018)’, there is an ambiguity. Please consider rephrasing the sentence.
-On Line 99, Please change the phrase ‘solubilize phosphates’ to ‘solubilized phosphates’.
-On Line 112, the sentence ‘Hence, their agricultural use as bioinoculants and the market focused on sustainable agriculture remains limited, accounting for only about 1% of the conventional agriculture market ' has ambiguity. Please consider rephrasing by separating the sentence.

Literature references are sufficient except:
In the materials and methods section (on line 121), If the isolates belongs to a previous study, it should be given as reference. it would be beneficial to give details on how the isolation procedure was done and how the authors identified Pseudomonas spp. Also, it would help the readers to understand if the authors mention the specific reason to choose sugarcane and agave plant to collect Pseudomonas spp.

Experimental design

Experimental results are clear and well defined with minor issues to address:
-Figure 5 is not discussed or referred in the manuscript. Please state the main findings which the readers should deduce from Fig. 5.
-In Fig 4, figure labels has some typos such as ‘Peudomonas’. Please check typing errors.
-Pseudomonas sp. C14 is seems to excel in every parameter authors characterized even exceeding mixed culture. It could boost the scientific storytelling if the authors provide a integrated matrix for scoring which one of the isolate is the most promising one. The evaluation should be discussed to comprehend possible reasons to observe the enhancement in growth parameters, while the other isolates and the mixed one failed to trigger the same.

Validity of the findings

Overall structure and validity of the findings are statistically defined. The connection of the outcomes with the literature is properly explained. Conclusion is within the scope of original research motivation. Unsupported statements are not present in the manuscript.

---

## Round 0.2 · Minor Revisions

Dear Dr. Luna-Cruz,

I am pleased to inform you that your manuscript is almost ready to be accepted for publication in PeerJ. The authors addressed all points raised by the reviewers, improving the manuscript.

As an Academic Editor, I only request a further change in the Title before publication. As stated by Rev.3, the first title is wordy…, but both are not clear.

I would like suggesting a third one: “Inoculation with Pseudomonas spp. in Solanum lycopersicum increases yield and fruit quality under nutrient shortage conditions”

In my opinion this title summarizes the main findings and the applicative value of your work.

In addition, a Section Editor noted the following issues that must also be addressed:

> There are substantial grammar problems evident in the abstract. I mostly stopped reading at that point and am returning this; the whole manuscript needs to be edited for proper use of language. examples: line 21 "absorption by t plants" (what is t?)
>
> +++ lines 23-26: "In this context, Plant Growth-Promoting Rhizobacteria (PGPR) offer agricultural advantages effects of five Pseudomonas isolates to achieve a 50% reduction in nitrogen, phosphorus, and calcium concentrations applied through hydroponic systems in the greenhouse cultivation of the El Cid F1 tomatoes ." This does not make sense; something omitted?
>
> +++ line 30 "phosphate and, and produce "
>
> +++ line 35 " Lycopene contend " . etc.

**Language Note:** The review process has identified that the English language must be improved. PeerJ can provide language editing services - please contact us at [email protected] for pricing (be sure to provide your manuscript number and title). Alternatively, you should make your own arrangements to improve the language quality and provide details in your response letter. – PeerJ Staff

---

## Round 0.3 · Minor Revisions

You have only edited the Abstract, but there are issues throughout the manuscript.

A few examples are shown below, but EVERYTHING NEEDS TO BE CHECKED, NOT JUST THE EXAMPLES IN THE LIST.

- line 66 "Under these conditions, the deficiency of even one nutrient its plant growth and development" (missing verb).
- line 75 "In addition, the recent rise in the price of fertilizer due to various factors (FAO, 2022) it is essential to emphasize the importance of..." (something missing here)
- line 105 "solubilized" (tense does not match the rest of the sentence).
- line 343 - 347. The results are repeated in this section "However, the difference widened at 130 days after planting. T50% of plants had on average, a height of 146 cm, stem diameter of 6.6 mm, and SPAD units (as an indirect measure of leaf nitrogen) at the beginning of the reproductive stage (Fig. 2). " and then "However, these differences became more pronounced by 130 days after transplanting. On average, plants under the T50% treatment reached a height of 146 cm, a stem diameter of 6.6 mm, and 40 SPAD units"

Please enlist the help of a fluent English speaker or use a professional editing service.

**Language Note:** The Academic Editor has identified that the English language must be improved. PeerJ can provide language editing services - please contact us at [email protected] for pricing (be sure to provide your manuscript number and title). Alternatively, you should make your own arrangements to improve the language quality and provide details in your response letter. – PeerJ Staff

---

## Round 0.4 · accepted · Accept

Dear Authors,

After reading the revised MS titled "Inoculation with Pseudomonas spp. in Solanum lycopersicum increases yield and fruit quality under nutrient shortage conditions", I think it could be accepted in the present form for publication.